# Ovarian Stimulation in Mice Resulted in Abnormal Placentation through Its Effects on Proliferation and Cytokine Production of Uterine NK Cells

**DOI:** 10.3390/ijms24065907

**Published:** 2023-03-21

**Authors:** Rong Ma, Ni Jin, Hui Lei, Jie Dong, Yujing Xiong, Chenxi Qian, Shuqiang Chen, Xiaohong Wang

**Affiliations:** Department of Obstetrics and Gynecology, Tangdu Hospital, Air Force Medical University, Xi’an 710038, China

**Keywords:** uNK, decidua, cytokines, DSCs, placentation, ER-α

## Abstract

Ovarian stimulation is associated with an increased incidence of abnormal placentation. Uterine natural killer (uNK) cells are the major subpopulation of decidual immune cells, which are crucial for placentation. In a previous study, we found that ovarian stimulation impairs uNK cell density on gestation day (GD) 8.5 in mice. However, it was not clear how ovarian stimulation led to a reduction in the density of uNK cells. In this study, we constructed two mouse models, an in vitro mouse embryo transfer model and an estrogen-stimulated mouse model. We used HE and PAS glycogen staining, immunohistochemical techniques, q-PCR, Western blot, and flow cytometry to analyze the mouse decidua and placenta, and the results showed that SO resulted in a fetal weight reduction, abnormal placental morphology, decreased placental vascular density, and abnormal density and function of uNK cells. Our results suggest that ovarian stimulation resulted in aberrant estrogen signaling and may contribute to the disorder of uNK cells caused by ovarian stimulation. Together, these results provide new insights into the mechanisms of aberrant maternal endocrine environments and abnormal placentation.

## 1. Introduction

Since 1978, more than 8 million infants worldwide have been delivered through assisted reproductive technologies (ARTs) [1]. However, increasing evidence shows that ART treatment is associated with increased adverse perinatal outcomes, including low birth weight, preterm labor, and preeclampsia, compared to those conceived naturally [2,3]. Ovarian stimulation plays a central role in ART, which enables the collection of more oocytes and thus more embryos per fresh cycle. A higher number of embryos increases the success rate of live births. However, compared with natural cycles, cycles involving ovarian stimulation are associated with a higher risk of adverse perinatal outcomes [4,5].

The placenta plays a critical role in controlling maternal–fetal resource allocation and mediating the intrauterine growth of the fetus [6]. Many adverse pregnancy outcomes, including preeclampsia, low birth weight, and preterm birth, arise from disorders of placental development [7,8,9]. Ovarian stimulation is associated with an increased incidence of abnormal placentation in humans [10,11]. In mice, studies demonstrated that superovulated environment postembryonic transfer significantly affects the distribution of cell types and vasculogenesis in the placenta [12,13]. These results suggest that placental abnormalities associated with ovarian stimulation may contribute to adverse perinatal outcomes and are responsible for the abnormal placentation seen during fresh in vitro fertilization (IVF) cycles.

In fresh autologous IVF cycles, the maternal endometrium is exposed to a supraphysiologic hormonal environment created through superovulation. Placental development requires fine-tuned crosstalk between the trophoblast cells and the maternal endometrium cells [14]. Uterine natural killer (uNK) cells are the major subpopulation of decidual immune cells, which secrete large amounts of cytokines/chemokines and angiogenic factors and regulate trophoblast invasion, spiral arterial modification, and placental formation [15]. The dysregulation of uNK cells has been associated with adverse perinatal outcomes, such as recurrent pregnancy loss, preeclampsia, and fetal growth retardation [16,17]. In a previous study, we found that ovarian stimulation impairs uNK cell density in the decidua basalis on gestation day (GD) 8.5 in mice [18]. However, it is not clear how ovarian stimulation led to a reduction in the density of uNK cells in decidua on GD 8.5 in mice. Determining the potential mechanisms by which ovarian stimulation impairs uNK cell density in the decidua is worthy of further study.

Decidualization is essential for embryo implantation and placental development. Decidualization involves the wholesale reprogramming of gene expression in endometrial stromal fibroblast cells, including growth factor, cytokine, and chemokine signaling [19]. Evidence indicates that decidualization accompanies the massive immune cell infiltration, including uNK cells [19]. Previous research has indicated that decidual stromal cells (DSCs) participate in the regulation of uNK cells by producing chemokines and cytokines, which have direct effects on uNK cell function [20]. The secretion of chemokines stimulated the recruitment of uNK cells from the maternal circulation [20,21]. Endometrium-derived cytokines have been shown to have direct effects on the differentiation and function of the uNK cells such that they acquire unique phenotypes in the uterine microenvironment [20]. Decidualization processes are orchestrated by ovarian steroid hormones through their cognate receptors [22]. In the uterine tissue of mammals, the predominant estrogen receptor (in both the myometrium and endometrium) is estrogen receptor α (ER-α). Uterine ER-α plays an essential role in decidualization [23]. In assisted reproduction treatment, multifollicular development is induced by hormonal stimulation to increase the number of embryos available for transfer. Ovarian stimulation inevitably increases the circulating serum estrogen level in patients [24]. The endometrial stromal cell is the first cell type that comes into contact with circulating estrogen in the endometrial capillaries: therefore, it represents the first line of response to estrogen. Estrogen binds to ER-α, resulting in receptor dimerization and nuclear translocation. The complex then binds to DNA target sequences to regulate gene expression [23]. The ablation of ERα in the uterus epithelium leads to a defect in stromal cell decidualization [25]. Ovarian stimulation treatment altered both the serum estrogen level and the ER-α expression level in the endometrium during the preimplantation period in mice [26]. We propose that the abnormal estrogen signal induced by ovarian stimulation may be related to the abnormal function of uNK cells.

During normal mouse pregnancy, abundant numbers of uNK cells differentiate at implantation sites and contribute to early, post-implantation endometrial angiogenesis and spiral arterial modification [27,28]. Mouse uNK cells can be identified using light microscopy in tissue sections stained with special protocols. Classically, mouse uNK cells were recognized as lymphocytes containing periodic acid–Schiff (PAS) reactive cytoplasmic granules. PAS reacts with cytoplasmic granules, rather than the uNK cell’s plasma membrane, so immature uNK cells are not identified by this reagent [29]. Dolichos biflorus (DBA) lectin is a lectin that specifically binds to terminal N-acetylga lactosamine (GalNAc). DBA lectin is a very selective and sensitive reagent for identifying mouse uNK cells, which are the only leukocytes in the uterus that bind to DBA [30,31]. DBA reactions, which stain not only the cytoplasmic granules but also the uNK cell membranes, have been widely adopted. No lymphocytes in any tissues of virgin mice or external to the uteri of pregnant mice have DBA lectin reactivity equivalent to that of uNK cells. Due to its identification of immature, agranular uNK cells histologically, its usefulness in the isolation of uNK cells for RNA isolation when conjugated to immunomagnetic beads, and its value in flow cytometry when directly or indirectly conjugated to fluorochromes [32], all mouse uNK cells can be identified using dual reagent histochemistry with DBA lectin and PAS [27,28]. Therefore, in the present study, we focused our attention on the effect of ovarian stimulation on PAS + DBA + uNK cells. We hypothesized that ovarian stimulation may indirectly affect the secretion of chemokines and cytokines during the early stage of pregnancy, thus affecting the recruitment, proliferation, and function of uNK cells in mice, which leads to abnormal placentation. To test this theory, we first analyzed the effects of ovarian stimulation on placental morphology, placental angiogenesis, and the proliferation activity and production of cytokines of uNK cells in the decidua basalis in mice. We further assessed the effects of ovarian stimulation on the production of chemokines and cytokines related to uNK functions. Furthermore, we evaluated the effect of a high estrogen level on the uNK cell density in the decidua basalis and the production of chemokines and cytokines that are related to uNK cell functions. Our findings demonstrate that ovarian stimulation impaired uNK cell density in the decidua basalis and is associated with the perturbation of estrogen signaling in mice.

## 2. Results

### 2.1. Ovarian Stimulation Resulted in Lower Fetal Weight on GD 18.5

We analyzed the impacts of ovarian stimulation on placental and fetal development on GD 18.5. We found that the fetal weights of the control and SO groups were 1523.74 ± 52.23 mg and 1207.84 ± 136.15 mg, respectively (Table 1). The fetal weights of the SO group were significantly lower than those of the control group (*p* < 0.001). There were no differences among the groups regarding placental weight (107.62 ± 3.33 mg vs. 101.63 ± 8.71 mg; *p* > 0.05; Table 1). However, the placental efficiency in the SO group (the ratio of fetal weight to placental weight) was reduced compared to that in the control group (14.17 ± 0.54% vs. 12.03 ± 2.12%, *p* < 0.01; Table 1). Thus, ovarian stimulation results in a lower fetal weight and placental efficiency on GD 18.5.

### 2.2. The Placental Morphology Was Perturbed by Ovarian Stimulation

To determine whether ovarian stimulation affects placental morphology at late gestation, we performed hematoxylin and eosin (HE) staining and PAS staining to examine GD 14.5 placentas. In all the groups, the spongiotrophoblast layer and labyrinth zone were distinguishable (Figure 1A). The labyrinth area/total area ratios of the control group and SO group were 0.47 ± 0.05 and 0.48 ± 0.05, and there was no difference between the two groups (Figure 1B).

We then used PAS staining to better define the defects in the spongiotrophoblast layer and labyrinth zone of the mouse placenta. The spongiotrophoblast layer was positive for PAS, which suggests that many of these spongiotrophoblasts were glycogen cells (Figure 1C). The PAS-positive area to total area was significantly higher in the placentas of the SO group than the control group (0.49 ± 0.05 vs. 0.55 ± 0.05; *p* < 0.05; Figure 1D).

### 2.3. Ovarian Stimulation Resulted in Lower Placental Angiogenesis on GD 14.5

Although there were no differences in placental weight among the two groups, we hypothesized that structural changes in the placenta may have contributed to the observed differences in fetal weight. We further assessed placental angiogenesis by calculating the placental MVD values at the labyrinth zone via the immunohistochemistry of placentas from SO (*n* = 5) and control (*n* = 5) recipients on GD 14.5. Immunohistochemical staining for laminin was used to measure the MVD values (Figure 2A–D). The placental MVD values in the control and SO groups were 329.33 ± 64.85 and 276.47 ± 39.37, respectively. The MVD values in the SO group were lower than those in the control group (*p* < 0.05; Figure 2E).

### 2.4. Effects of Ovarian Stimulation on the Uterine Natural Killer Cells in the Decidual Basalis at Different Gestational Ages

We analyzed the impacts of ovarian stimulation on uNK cell density by PAS staining and DBA lectin histochemical analyses in the decidua basalis (DB) on GD 6.5, GD 9.5, and GD 12.5. The PAS + DBA + uNK cell densities (the ratios of the PAS + DBA + uNK positive cells to the total cells) of the control and SO groups were compared at three time points (Figure 3). Notably, the density of PAS + DBA + uNK cells in DB areas significantly declined in SO pregnant uteri compared with control pregnant uteri on GD 6.5 (Figure 3A,B,G), GD 9.5 (Figure 3C,D,H), and GD 12.5 (Figure 3E,F,I). In mice, mature uNK cells are large, granule-rich cells, and immature uNK cells are characterized by low numbers of cytoplasmic granules. It was found that most of the uNK cells in SO pregnant uteri were immature compared with the control group on GD 9.5 (Figure 3C,D) and GD 12.5 (Figure 3E,F).

### 2.5. Effects of Ovarian Stimulation on the Function of Uterine Natural Killer Cells in the Decidual Basalis

As a reduced PAS + DBA + uNK cell density was found in the decidua basalis of the SO group, we evaluated the frequency of subtypes of the Ki67 index in the control and SO groups. Compared with the control group (63.27 ± 7.62%), a lower level of Ki67 expression was detected for CD45 + CD3 − DBA + uNK cells in the SO group (51.18 ± 5.06%, *p* < 0.05; Figure 4A,E). Comparing the cytokine profiles of the CD45 + CD3 − DBA + uNK cell subset between the control and SO groups demonstrated that the percentage of IFN-γ-producing cells was significantly lower in the SO mice (6.56 ± 1.05%) as compared with control mice (10.39 ± 2.06%, *p* < 0.01; Figure 4B,F), and the percentages of VEGF- and PLGF-producing cells in the SO groups (26.86 ± 3.80%, 28.07 ± 5.75%, respectively) were lower than in control mice (43.09 ± 10.83%, *p* < 0.05. and 38.71 ± 7.45%, *p* < 0.05; Figure 4C,D,G,H). Our findings indicated that SO changed the cytokine profile of the CD45 + CD3 − DBA + uNK cell subset.

### 2.6. Key Chemokines and Cytokines That Regulated the Recruitment, Proliferation, and Function of uNK Cells in GD 9.5 Decidual Tissue Were Downregulated by Ovarian Stimulation

Chemokine C-X-C ligand 10 (CXCL10), CXCL11, CXCL12, CXCL14, CX3C chemokine ligand 1 (CX3CL1), and CC chemokine ligand 2 (CCL2) in decidual tissue are crucial for uNK cell recruitment. It was found that the relative expression levels of CCL2, CXCL10, CXCL11, CXCL12, and CX3CL1 in decidual tissue were lower in the SO group than in the control group (*p* < 0.05; Figure 5A). However, there were no changes in the expression of CXCL14 between the two groups. IL-15, IL-24, placenta growth factor (PLGF), and TGF-β have direct effects on the differentiation and function of uNK cells. It was found that the relative expression levels of IL-15, IL-24, PLGF, and TGF-β in decidual tissue were lower in the SO group than in the control group (*p* < 0.05; Figure 5B).

### 2.7. Aberrant Estrogen Signaling May Contribute to the Disorder of uNK Cells Related to Ovarian Stimulation

We determined whether ovarian stimulation can influence ER-α transcription activity in mice. As shown in Figure 6A–C, ovarian stimulation markedly induced ER-α transcription activity in GD 9.5 decidual tissue compared with the control group (*p* < 0.05). To determine whether high estrogen levels could have a direct impact on uNK cell density in the decidua basalis, the pregnant mice were treated with either estrogen (Sigma) dissolved in corn oil or an equal volume of pure corn oil from GD 5.5 to GD 8.5. Treatment with estrogen significantly reduced the PAS + DBA + uNK cell density in the decidua basalis (Figure 6D,E; *p* < 0.001).

### 2.8. High Estrogen Levels Suppressed the Expression of Chemokines and Cytokines That Regulate the Functions of uNK Cells in Mouse Decidual Stromal Cells

The purity of isolated endometrial stromal cells was determined by immunofluorescence staining of vimentin and cytokeratin. Over 95% of the isolated cells were identified as vimentin-positive and cytokeratin-negative stromal cells (Appendix A). We also validated the in vitro decidualization by detecting the expression of dPRP, a well-known decidualization marker of endometrial stromal cells. The expression of dPRP was significantly up-regulated when estrogen and progesterone were added (10 nM E2 and 1 μM P4) (*p* < 0.05; Appendix A). In order to evaluate the high estrogen effects on the expression of chemokines and cytokines that regulated the functions of uNK cells, decidual stromal cells were stimulated with different estrogen concentrations for 72 h. The relative expression levels of CXCL12, CXCL14, and CX3CL1 were reduced after high-concentration E2 treatment (*p* < 0.05; Figure 7A). Moreover, the relative expression levels of IL-15, PLGF, and TGF-β in decidual stromal cells were lower in the high-estrogen group than in the control group (*p* < 0.05; Figure 7B).

## 3. Discussion

In human assisted reproductive technology, ovarian stimulation protocols use follicle-stimulating hormone (FSH) in combination with gonadotropin-releasing hormone (GnRH) (agonists or antagonists), as oral supplements [33]. Specific ovarian stimulation programs vary from individual to individual and are generally divided into three types: the gonadotropin-releasing hormone long program, the gonadotropin-releasing hormone short program, and the antagonist program [11,34]. Ovarian stimulation commonly involves the use of hCG: multi-follicular development is induced using exogenous gonadotrophins, and hCG is injected in order to trigger oocyte maturation before retrieving oocytes to perform in vitro fertilization (IVF) or intracytoplasmic sperm injection (ICSI). Ovarian stimulation has been associated with an increased risk of abnormal placentation, leading to increased adverse perinatal outcomes compared to those conceived naturally [4,5]. Animal studies have confirmed that the superovulated environment post-embryonic transfer significantly affects the distribution of cell types and vasculogenesis in the placenta [12,13]. In the mouse model, the general protocol of ovarian stimulation is that female mice are intraperitoneally injected with a pregnant mare’s serum gonadotrophin (PMSG), followed by the same international units of hCG 48 h later [13,35,36]. We used a novel mouse model to examine how the ovarian-stimulation-induced unphysiological uterine environment affects fetal growth and placental development. Our results indicate that superovulation resulted in abnormal placentation and is associated with reduced proliferation activity and the production of cytokines including IFN-γ, VEGF, and PLGF of CD45 + CD3 − DBA + uNK cells in the decidua basalis. Additionally, we have demonstrated that the production of chemokines and cytokines related to uNK recruitment, proliferation, and differentiation in the decidua is suppressed by ovarian stimulation.

Consistent with previous findings, in the precent study, we also found that the ovarian-stimulation-induced unphysiological post-implantation environment caused a lower fetal weight and is associated with an abnormal placental morphology and lower placental angiogenesis [12,13]. Our laboratory previously found that ovarian stimulation led to a reduction in the density of DBA + uNK cells in the decidua on GD 8.5 in mice [18]. Here, we analyzed the impacts of ovarian stimulation on uNK cell density by PAS staining and DBA lectin histochemical analyses in the decidua basalis on GD 6.5, GD 9.5, and GD 12.5, which are crucial periods for placentation. We demonstrate that ovarian stimulation reduced the density of PAS + DBA + uNK cells in the decidua basalis at the three time points. We also found that the proliferation activity and the production of cytokines including IFN-γ, VEGF, and PLGF of PAS + DBA + uNK cells were affected by ovarian stimulation. These results suggest that ovarian stimulation leads to a decrease in the proliferation activity and the production of cytokines of uNK cells during the whole early gestation period, which is crucial for placental development.

Decidualization refers to the differentiation of endometrial stromal fibroblast cells into specialized secretory decidual stromal cells [37]. During decidualization, the endometrial stromal cells exhibit morphological and biochemical alterations to transform into a highly secretory phenotype [38]. Decidualization involves the wholesale reprogramming of gene expression in human endometrial stromal cells, including growth factor, cytokine, and chemokine signaling [19]. Evidence indicates that decidualization accompanies the massive immune cell infiltration, including uNK cells [19]. Previous research has indicated that DSCs participate in the regulation of uNK cells by producing chemokines and cytokines, which have direct effects on the recruitment, proliferation, and differentiation of uNK cells [20]. Evidence suggests that CX3CL1, CXCL10, CXCL11, CXCL12, and CXCL14 control the recruitment of NK cells in the decidua [39,40,41]. In the present study, we found that the expression of CX3CL1, CXCL10, CXCL11, and CXCL12 in decidual tissue was significantly lower in the SO group than in the control group. Evidence indicated that cytokines provided by DSCs control uterine NK cell development and functional differentiation. IL-15 is produced by the DSCs induced by progesterone and drives uNK cell proliferation and differentiation [42]. IL-24 secreted by DSCs promotes the differentiation of uNK with high levels of inhibitory receptors, immunotolerance, and angiogenic cytokines [29]. PLGF plays an important role in successful uNK cell proliferation and/or differentiation in mice [43].

Estrogen and progesterone are the major mediators of decidualization [19]. The proliferation, secretion, and decidualization of endometrial stromal cells are regulated by estrogen and progesterone [41]. It was reported that uterine ER-α plays an essential role in decidualization, and that the ablation of ER-α in the uterus epithelium leads to a defect in stromal cell decidualization [25]. Studies identified that the de novo synthesis of estrogen in pregnant uteri is critical for stromal decidualization and angiogenesis [44]. There is evidence showing that estrogen can regulate the recruitment, proliferation, differentiation, and function of uNK cells through direct means or indirect ones [41]. HAND2, a transcription factor in the response to estradiol and progesterone, directly regulates IL-15 expression in human endometrial stromal cells [45]. Subsequent experiments have shown that estrogen can increase the expression of CXCL10 and CXCL11 in endometrial cells, which serve to regulate the recruitment of NK cells in the decidua [39]. Ovarian stimulation treatment altered both the serum estrogen level and the ER-α expression level in the endometrium during the preimplantation period in mice [26]. In the present study, it was found that ovarian stimulation markedly induced ER-α transcription activity in GD 9.5 decidual tissue compared with the control group. To determine whether high estrogen levels could have a direct impact on uNK cell density and maturity in the decidua basalis, the pregnant mice were treated with estrogen. It was found that the uNK cell density and maturity were significantly reduced by the high estrogen levels in mice. We found that the mouse endometrial stromal cells treated with higher estrogen downregulated the expression of chemokines and cytokines, including CXCL12, CXCL14, CX3CL1, IL-15, PLGF, and TGF-β, which were also downregulated by ovarian stimulation in the decidua. These results suggested that abnormal estrogen signaling induced by superovulation may affect the recruitment, proliferation, and differentiation of the uNK cells by downregulating the expression of chemokines and cytokines. In the present study, we demonstrated that the maternal abnormal estrogen signaling induced by superovulation might impair uNK cell accumulation and maturity. This provides a new insight into the mechanisms of aberrant maternal endocrine environments and abnormal placentation.

## 4. Materials and Methods

### 4.1. Animals

Virgin CD1 female mice (8–10 weeks of age) and adult CD1 male mice were used. Around 240 female mice and 25 male mice were used in the study. All animals were provided with nesting material and housed in cages maintained under a constant 12-h light/12-h dark cycle at 21–23 °C, with free access to standard chow and tap water. The present study was approved by the Medical Ethics Committee of Tangdu Hospital at the Air Force Medical University (Permit Number: 202203-27).

### 4.2. Embryo Transfer

To obtain blastocysts for transfer, female mice were mated with males, and mating was confirmed by the presence of a copulatory plug on the morning following mating (0.5 days post-coitum). On post-coital day 3.5, the blastocysts were flushed from the uterine horns. The blastocysts were transferred within 1 h. Two groups of pseudopregnant CD1 females were obtained: the natural mating group (control group) and the ovulation-stimulating hormone group (SO group). The females in the SO group were superovulated by intraperitoneal injection of 5 international units (IU) of PMSG (hor-272-a, ProSpec, Ness-Ziona, Israel) followed by 5 IU of hCG (230734, Millipore, Billerica, MA, USA) 48 h later. The superovulated females were mated with vasectomized males soon after the injection of hCG. Control recipients were obtained by mating CD1 females with vasectomized males. Mating was confirmed by the presence of a copulatory plug. On post-coital day 3.5, 8 blastocysts were transferred into a single uterine horn of each recipient according to standard procedures.

### 4.3. Placenta Dissection

Fetuses and placentae were harvested on GD 14.5 or GD 18.5. Pregnant mice were euthanized by anesthesia followed by cervical dislocation. After dissection, the placental and fetal wet weights were recorded, and each placenta was immediately preserved. Some were fixed overnight in 4% paraformaldehyde and embedded in paraffin wax for histological analyses. Standard procedures were used for automated tissue processing and paraffin embedding. Serial sections were prepared at 5 µm thickness.

### 4.4. Placental HE and PAS Staining

HE staining was used to observe the placenta morphology. The placenta was fixed in 4 % paraformaldehyde, conventionally dehydrated and embedded, sliced, and roasted at 60 °C for 2 h, and then conventionally dewaxed and stained with hematoxylin for 2 min. After differentiation with an acid differentiation solution and eosin staining, it was conventionally dehydrated, made transparent, and sealed.

Periodic acid–Schiff (PAS) staining was also used to observe the placenta morphology. Sections were conventionally dewaxed and were incubated with oxidants at room temperature for 5 min. Sections were incubated with Schiff reagent for 20 min, rinsed with running water for 10 min, and stained with hematoxylin for 2 min, and then dehydrated, made transparent, and sealed.

### 4.5. Immunohistochemistry for Determination of Placental Microvascular Density

The placenta slices were subjected to routine dewaxing, hydration, and the blocking of endogenous peroxidase activity. Based on the instructions of the immunohistochemistry kit, the slices were incubated overnight with 1:100 dilutions of primary antibodies against laminin (ab11575, Abcam, Cambridge, UK) and were then incubated with Polymer Helper and horseradish peroxidase-labeled secondary antibodies (1:20,000, ab205718, Abcam, Cambridge, UK). DAB solution was used for color development. Cells with positive expression of laminin exhibited brownish-yellow particles. PBS was used instead of a primary antibody in the negative control group. The expression of laminin in cells was observed under a microscope. The microvascular densities (MVD) values of laminin-positive cells were determined under 400× magnification in five randomly selected fields. The mean values of the five fields were used as the MVD values.

### 4.6. DBA Lectin and PAS Dual Staining

The decidua tissue of mice was fixed with 4% paraformaldehyde, embedded by conventional dehydration, sliced, dewaxed, and hydrated conventionally. Antigen was retrieved by citric acid buffer (PH6.0) microwave antigen retrieval. Endogenous peroxidase activity was blocked by incubating the sections in 3% peroxide in methanol for 30 min at room temperature. After three washes in PBS, we blocked nonspecific binding by incubating sections with carbon-free blocking solution (SP-5040, VECTOR, Burlingame, CA, USA) for 30 min at room temperature, followed by incubation with biotin-conjugated DBA lectin (1:1200, L6533, Sigma, St. Louis, MO, USA) overnight at 4 °C. Sections were incubated with horseradish enzyme-labeled streptomycin working solution at room temperature for 15 min before DBA staining. PAS staining was performed with standard practice. Sections were counterstained with hematoxylin and observed by light microscopy.

### 4.7. Isolation and In Vitro Decidualization of Mouse Endometrial Stromal Cells

The isolation and culture of mouse endometrial stromal cells were performed using previously described methods [46]. In brief, mice at 6–8 weeks of age were injected subcutaneously with 100 µL of the 17β-estradiol (E2) solution (100 ng/100 µL) for 3 consecutive days. We euthanized the animals by cervical dislocation and cut out the horn to isolate the mouse endometrial stromal cells. The cells collected with the DMEM/F12 complete medium were passed through a 40 μm sieve to collect single cells with 500× *g* centrifugation for 7 min. The isolated stromal cells were further cultured in fresh complete medium at 37 °C with 5% CO_2_ before treatments. These cells were respectively treated with 1 μM of progesterone and different concentrations of 17β-estradiol (100 nM and 1000 nM, respectively) to induce in vitro decidualization. We analyzed the influence of different concentrations of 17β-estradiol on the expression of chemokines and cytokines.

### 4.8. RNA Extraction, cDNA Preparation, and Real-Time PCR Analysis

A pool of six samples obtained from at least two litters from the same group was used for total RNA extraction. RNA was extracted with TRIzol (Invitrogen Life Technologies, 15596018) according to the manufacturer’s instructions and treated with DNase I to eliminate genomic DNA contamination. Reverse transcription (RT) was accomplished using a commercially available first-strand cDNA synthesis kit (RR036A, TakaRa, Tokyo, Japan). The RT reactions were performed on 1 µg of total RNA, following the manufacturer’s protocol. Real-time PCR was performed with a Bio-Rad CFX96 real-time PCR instrument (Bio-Rad, Hercules, CA, USA) and TB Green™ Advantage^®^ qPCR Premix for Q-PCR, according to the manufacturer’s protocol (639676, TakaRa, Tokyo, Japan). Gene expression was normalized using the housekeeping gene *Gapdh* as the reference gene. The primer sequences for the genes analyzed are listed in Appendix A. The primer sequences were obtained from PrimerBank or designed using Primer Express. The samples were analyzed using the ΔΔCt method.

### 4.9. Western Blot

Decidual tissue was homogenized on ice in tissue protein extraction reagent (78510, Thermo, Waltham, MA, USA) with proteinase inhibitors (HY-K0010, MCE, Monmouth Junction, NJ, USA) and phosphatase inhibitors (4906837001, Roche, Basel, Switzerland). The protein lysate was placed on ice for 20 min and centrifuged at 14,000 rpm for 20 min at 4 °C. The extracted supernatants were then harvested for subsequent testing. The protein concentration was detected using a BCA protein assay kit (23227, Thermo, Waltham, MA, USA). Proteins were separated using sodium dodecyl sulphate Tris-glycine gels (Bio-Rad), transferred onto PVDF membranes, blocked in 5% non-fat milk, and hybridized overnight at 4 °C with primary antibodies. The primary antibodies included ER-α (1:1000, ab32063, Abcam, Cambridge, UK) and Actin (1:1000, 4970S, CST, Danvers, MA, USA). The secondary antibody that we used was HRP-labeled goat anti-rabbit IgG (1:3000, 7074P2, CST, Danvers, MA, USA). The membranes were washed with Tris-buffered saline containing 0.01% Tween-20. Protein bands were visualized using Immobilon Chemiluminescence Reagent (WBKLS0500, Millipore, Billerica, MA, USA) and analyzed using a ChemiDox Gel imaging system (Bio-Rad, Hercules, CA, USA).

### 4.10. Flow Cytometry (FCM)

Females at GD 8.5 were euthanized and the decidua samples were washed and minced into small pieces. Decidual lymphocytes were released by digesting the tissues with 1 mg/mL collagenase type IV (02195110-CF, MP, Irvine, CA, USA) and 0.01 mg/mL DNase I (0219006210, MP, Irvine, CA, USA) in RPMI 1640 medium (10-040-CVR, Corning, NY, USA) for 60 min at 37 °C. The suspensions were strained through a nylon mesh (70 μm) and were collected by centrifugation and resuspended in 42% Percoll solution and added to the upper part of 70% Percoll separation solution, with 1260× *g* at room temperature, for 30 min, to obtain mouse uterine tissue mononuclear cells. Viable cells were counted and then immediately used for subsequent flow cytometry analysis. In addition, CD45-APC/Cy7, CD3-FITC, DBA-PE, Ki67-PE/Cy7, IFN-γ-APC (BioLegend, San Diego, CA, USA) and CD45-Percp/Cy5.5, CD3-FITC, DBA-PE, VEGF-APC/Cy7, PLGF-APC (BioLegend, San Diego, CA, USA) monoclonal antibodies were used for the detection of Ki67 and intracellular cytokines. Flow cytometric analysis was performed via nine-color flow cytometry using the Novo Express software 1.4.1. In every sample, 5 × 10^4^ cells were analyzed.

### 4.11. Statistical Analysis

Quantitative data are presented as the means ± standard deviations (SDs). For the fetal weight and placental weight data, we reanalyzed the associated data in the unit of the litter; statistics were calculated using the mean value for each foster mother, and therefore “*n*” represents the number of litters. The differences between the control and SO groups were determined statistically using Student’s *t*-test. All the analyses were performed using Prism8.0. Results were considered to be statistically significant if *p* < 0.05. For all analyses, *p* > 0.05; * *p* < 0.05; ** *p* < 0.01, and *** *p* < 0.001.

## 5. Conclusions

The results for our study showed that ovarian stimulation may cause abnormal estrogen signaling. The increased estrogen levels after ovarian stimulation suppressed the secretion of chemokines and cytokines by upregulating the expression of ER-α in decidual stromal cells. This decrease in chemokines and cytokines led to the abnormal proliferation and differentiation of uNK cells, which further led to the abnormal morphology and function of the placenta. In summary, we found that ovarian stimulation resulted in aberrant estrogen signaling and may contribute to the disorder of uNK cells caused by ovarian stimulation (Figure 8). Together, these results provide new insights into the mechanisms of aberrant maternal endocrine environments and abnormal placentation.

## Figures and Tables

**Figure 1 ijms-24-05907-f001:**
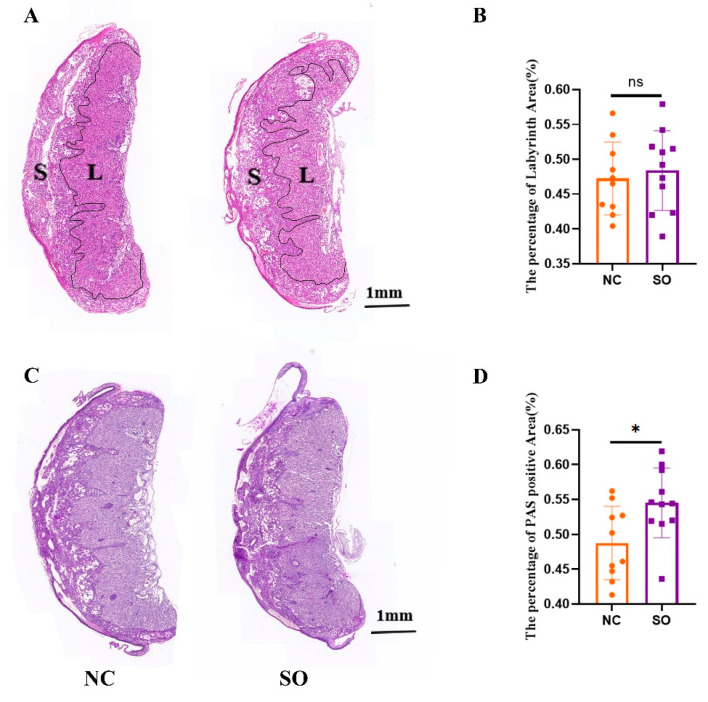
The placental morphology was perturbed by ovarian stimulation. (**A**) The spongiotrophoblast layer and labyrinth zone were distinguishable by HE staining. (**B**) There was no difference in the labyrinth area/total area ratios between the control group (*n* = 10) and SO group (*n* = 11). Data are presented as the means ± SD and were evaluated using Student’s *t*-test. (**C**) The spongiotrophoblast layer and labyrinth zone of the mouse placenta were better defined by PAS staining. The spongiotrophoblast layer was positive for PAS, which suggested that many of these spongiotrophoblasts were glycogen cells. (**D**) The PAS-positive area to total area was significantly higher in the placentas of the SO group (*n* = 11) than the control group (*n* = 10). Different colors represent different groups: orange indicates the control group; purple indicates the SO group. Data are presented as the means ± SD and were evaluated using Student’s *t*-test. ns, not significant. * *p* < 0.05.

**Figure 2 ijms-24-05907-f002:**
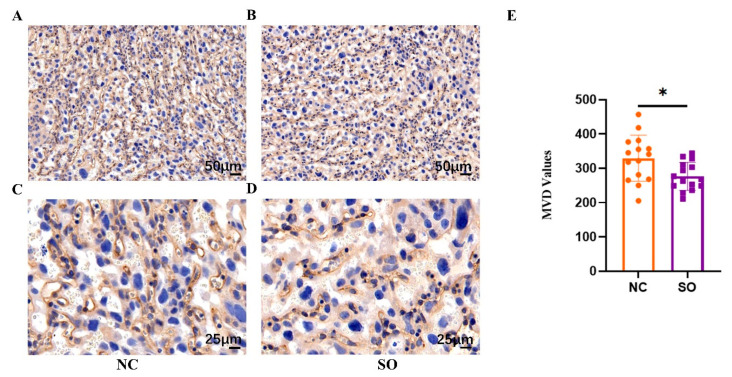
Ovarian stimulation resulted in lower placental angiogenesis on GD 14.5. (**A**,**B**) Immunohistochemical staining for laminin was used to measure the placental MVD values in the control and SO groups. Scale bars: 25 μm. (**C**,**D**) Immunohistochemical staining for laminin was used to measure the placental MVD values in the control and SO groups. Scale bars: 25 μm. (**E**) The MVD values in the SO group were lower than those in the control group. Different colors represent different groups: orange indicates the control group; purple indicates the SO group. Data are presented as the means ± SD and were evaluated using Student’s *t*-test (*n* = 15). * *p* < 0.05.

**Figure 3 ijms-24-05907-f003:**
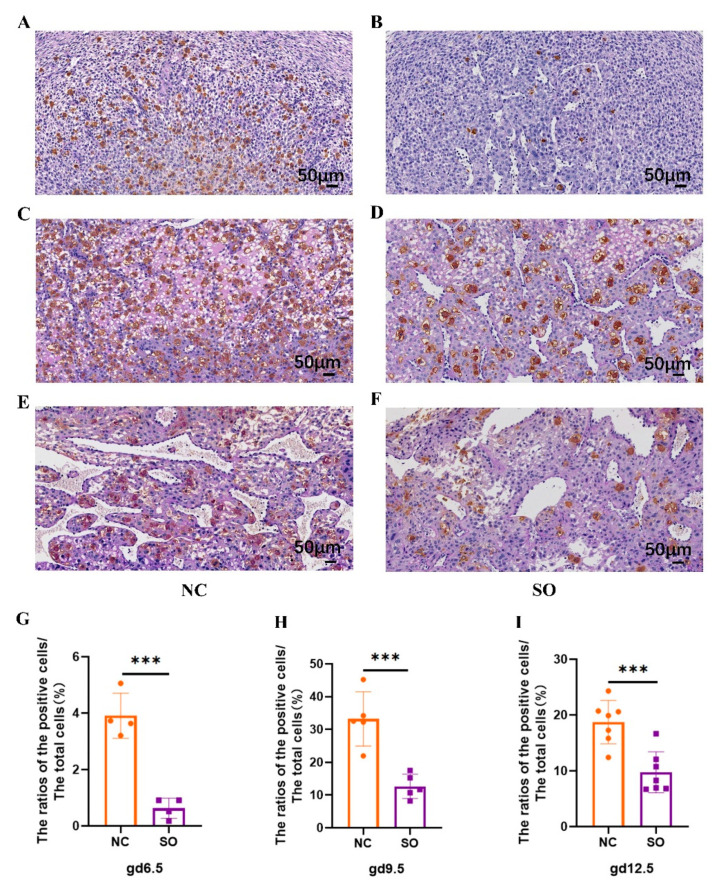
Effects of ovarian stimulation on the uterine natural killer cells in the decidual basalis at different gestational ages. (**A**,**B**) PAS staining and DBA lectin histochemical analyses for uNK cell density in the decidua basalis on GD 6.5. Scale bars: 50 μm. (**C**,**D**) PAS staining and DBA lectin histochemical analyses for uNK cell density in the decidua basalis on GD 9.5. Scale bars: 50 μm. (**E**,**F**) PAS staining and DBA lectin histochemical analyses for uNK cell density in the decidua basalis on GD 12.5. Scale bars: 50 μm. (**G**) The density of PAS + DBA + uNK cells in the decidual basalis area significantly declined in SO pregnant uteri compared with control pregnant uteri on GD 6.5 (*n* = 4). (**H**) The density of PAS + DBA + uNK cells in the decidual basalis area significantly declined in SO pregnant uteri compared with control pregnant uteri on GD 9.5 (*n* = 5). (**I**) The density of PAS + DBA + uNK cells in the decidual basalis area significantly declined in SO pregnant uteri compared with control pregnant uteri on GD 12.5 (*n* = 7). Different colors represent different groups: orange indicates the control group; purple indicates the SO group. Data are presented as the means ± SD and were evaluated using Student’s *t*-test. *** *p* < 0.001.

**Figure 4 ijms-24-05907-f004:**
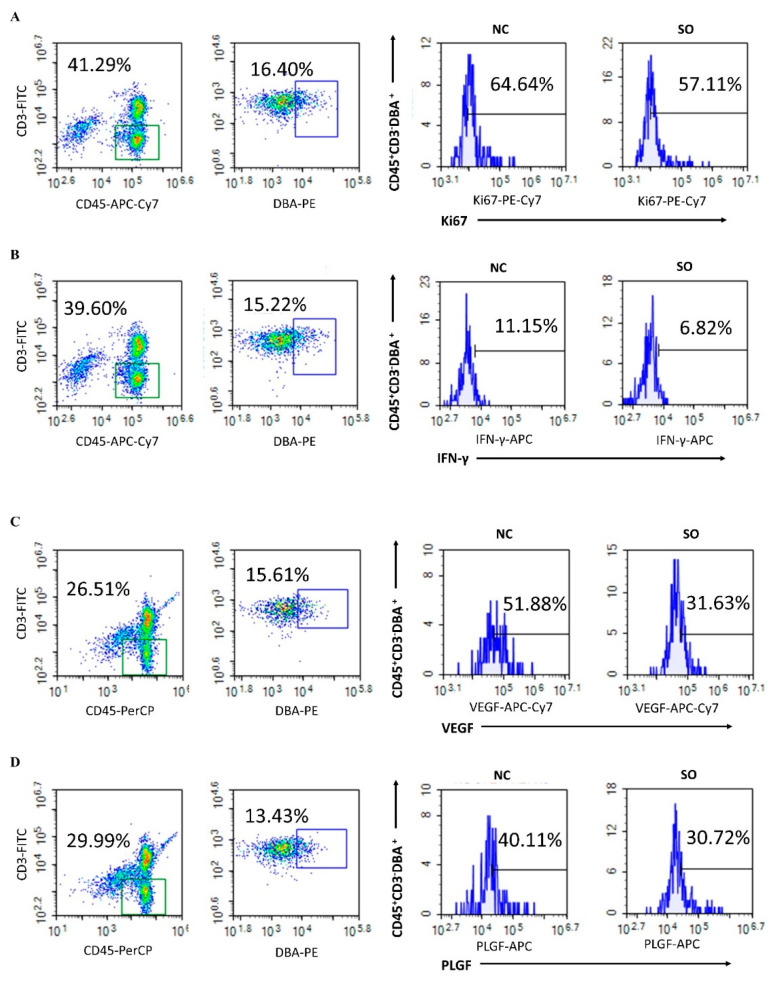
Effects of ovarian stimulation on the function of uterine natural killer cells in the decidual basalis. (**A**) The frequency of subtypes of Ki67 in CD45 + CD3 − DBA + uNK cells between the control and SO groups. (**B**) The percentage of IFN-γ in CD45 + CD3 − DBA + uNK cells between the control and SO groups. (**C**) The percentage of VEGF in CD45 + CD3 − DBA + uNK cells between the control and SO groups. (**D**) The percentage of PLGF in CD45 + CD3 − DBA + uNK cells between the control and the SO groups. (**E**) Compared with the control group, Ki67 expression was lower in the SO group. (**F**) The percentage of IFN-γ was significantly decreased in the SO group compared with the control group. (**G**) The percentage of VEGF was significantly decreased in the SO group compared with the control group. (**H**) The percentage of PLGF was significantly decreased in the SO group compared with the control group. Different colors represent different groups: orange indicates the control group; purple indicates the SO group. Data are presented as the means ± SD and were evaluated using Student’s *t*-test (*n* = 6). * *p* < 0.05 and ** *p* < 0.01.

**Figure 5 ijms-24-05907-f005:**
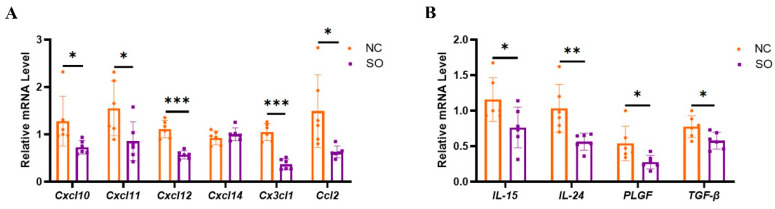
Key chemokines and cytokines that regulated the recruitment, proliferation, and function of uNK cells in GD 9.5 decidual tissue were downregulated by ovarian stimulation. (**A**) RT-PCR analysis of chemokines’ RNA expression in GD 9.5 decidual tissue. (**B**) RT-PCR analysis of cytokines’ RNA expression in GD 9.5 decidual tissue. Different colors represent different groups: orange indicates the control group; purple indicates the SO group. Data are presented as the means ± SD and were evaluated using Student’s *t*-test (*n* = 6). * *p* < 0.05, ** *p* < 0.01 and *** *p* < 0.001.

**Figure 6 ijms-24-05907-f006:**
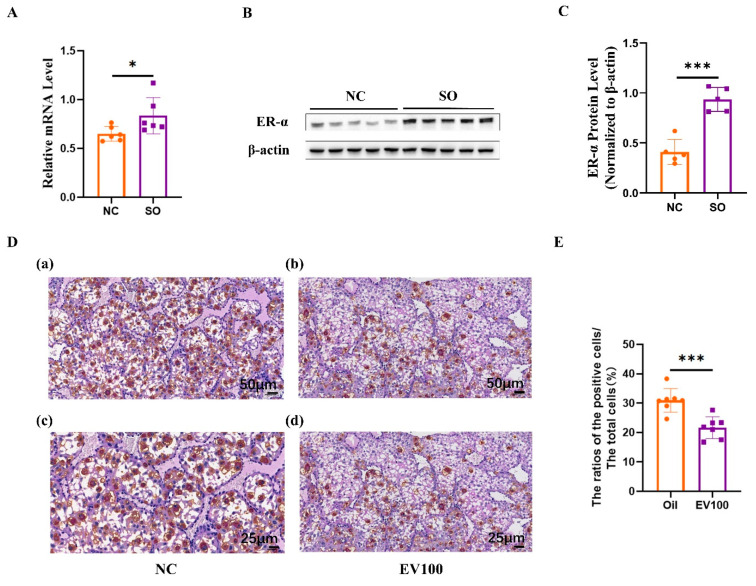
Aberrant estrogen signaling may contribute to the disorder of uNK cells related to ovarian stimulation. (**A**) RT-PCR analysis of ER-α RNA expression in GD 9.5 decidual tissue (n = 6). (**B**,**C**) Western blotting analysis showing elevated expression of ER-α in SO group in GD 9.5 decidual tissue compared with the control group (n = 5). (**D**) (**a**,**b**): PAS staining and DBA lectin histochemical analyses for uNK cell density in the decidua basalis on GD 9.5 between the oil group and EV100 group. Scale bars: 50 μm. (**c**,**d**): PAS staining and DBA lectin histochemical analyses for uNK cell density in the decidua basalis on GD 9.5 between the oil group and EV100 group. Scale bars: 25 μm. (**E**) The density of PAS + DBA + uNK cells in the decidual basalis area significantly declined in the EV100 group compared with the oil group (n = 7). Different colors represent different groups. Data are presented as the means ± SD and were evaluated using Student’s *t*-test. * *p* < 0.05, and *** *p* < 0.001.

**Figure 7 ijms-24-05907-f007:**
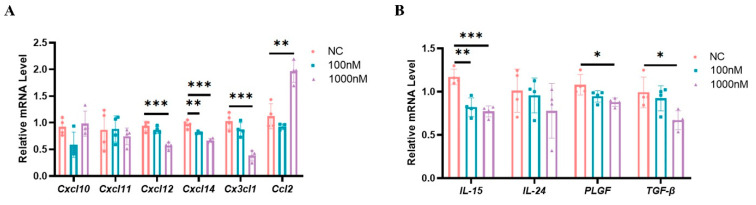
High estrogen levels suppressed the expression of chemokines and cytokines that regulate the functions of uNK cells in mouse decidual stromal cells. (**A**) RT-PCR analysis of chemokines’ RNA expression in mouse decidual stromal cells. (**B**) RT-PCR analysis of cytokines’ RNA expression in mouse decidual stromal cells. Different colors represent different groups. Data are presented as means ± SD and were evaluated using Student’s *t*-test (*n* = 4). * *p* < 0.05, ** *p* < 0.01 and *** *p* < 0.001.

**Figure 8 ijms-24-05907-f008:**
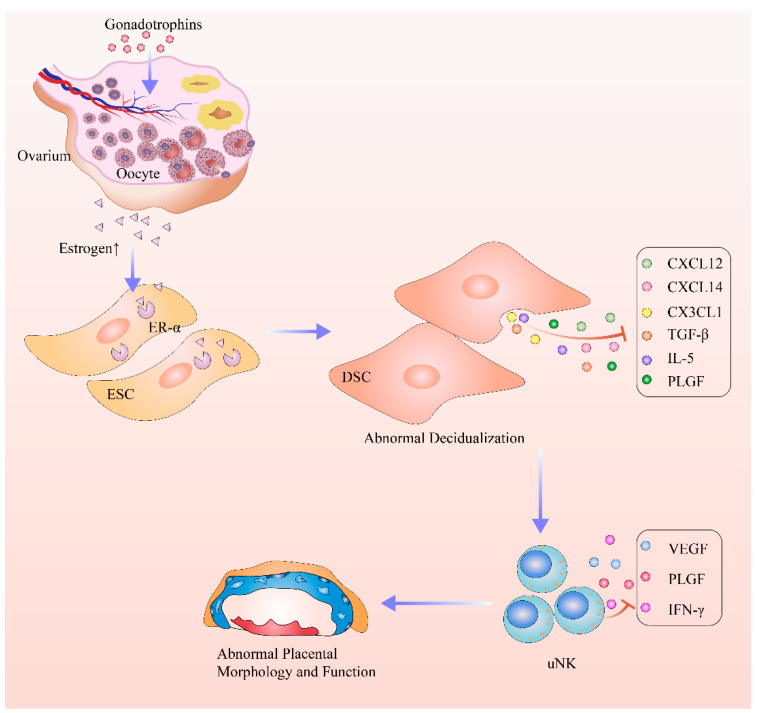
Mechanisms of aberrant maternal endocrine environments and abnormal placentation.

**Table 1 ijms-24-05907-t001:** Litter size, placental weight, fetal weight, and placental efficiency.

Gestational Age	Group	Litters	Litter Size	Placental Weight (mg)	Fetal Weight (mg)	Placental Efficiency (%)
GD 18.5	NC	10	10.40 ± 0.80	107.62 ± 3.33	1523.74 ± 52.23	14.17 ± 0.54
	SO	10	10.00 ± 2.24	101.63 ± 8.71	1207.84 ± 136.15	12.03 ± 2.12
	*p*		0.620	0.070	<0.001	<0.01

Data are presented as means ± SDs.

## Data Availability

The data presented in this study are contained within the article and the Appendix A.

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
