# Peer review of "Ovarian Stimulation in Mice Resulted in Abnormal Placentation through Its Effects on Proliferation and Cytokine Production of Uterine NK Cells"

_ijms, 2023, doi:10.3390/ijms24065907_

Round 1
Reviewer 1 Report
This article is devoted to the interesting topic of the effect of ovarian stimulation, which is often used in HRT programs, on the course of pregnancy and on the development of the placenta.
The article has an original design and is built according to an interesting plan. The discussion of the results is detailed. The description of the Materials and Methods is detailed.
Question: Why were uNK taken to estimate the population DBA lectin and PAS?
According to the literature, the population of natural killer cells is heterogeneous [Chen Z, Zhang J, Hatta K, Lima PD, Yadi H, Colucci F, Yamada AT, Croy BA. DBA-lectin reactivity defines mouse uterine natural killer cell subsets with biased gene expression. Biol Reprod. 2012 Oct 4;87(4):81. doi: 10.1095/biolreprod.112.102293.] Why did you bring these particular markers?
The article contains a sufficient number of figures and tables.
Author Response
We appreciate very much your careful review, constructive comments, and kind corrections to our manuscript. We have carefully revised the manuscript according to your comments, and we have incorporated all corrections marked with underlined RED font in the revised version. We hope that we have made some progress in this version.
This article is devoted to the interesting topic of the effect of ovarian stimulation, which is often used in HRT programs, on the course of pregnancy and on the development of the placenta.
The article has an original design and is built according to an interesting plan. The discussion of the results is detailed. The description of the Materials and Methods is detailed.
The article contains a sufficient number of figures and tables.
Point 1: Why were uNK taken to estimate the population DBA lectin and PAS?
According to the literature, the population of natural killer cells is heterogeneous [Chen Z, Zhang J, Hatta K, Lima PD, Yadi H, Colucci F, Yamada AT, Croy BA. DBA-lectin reactivity defines mouse uterine natural killer cell subsets with biased gene expression. Biol Reprod. 2012 Oct 4;87(4):81. doi: 10.1095/biolreprod.112.102293.] Why did you bring these particular markers?
Response 1: Thank you for your question and kindly advice. During normal mouse pregnancy, abundant numbers of uterine natural killer (uNK) cells differentiate at implantation sites and contribute to early, post-implantation endometrial angiogenesis and to spiral arterial modification[1]. Mouse uNK cells are confidently recognized by light microscopy in tissue sections stained with special protocols. Classically, mouse uNK cells were identified as lymphocytes containing Periodic Acid Schiff's (PAS) reactive cytoplasmic granules. PAS reacts with cytoplasmic granules, not with the uNK cell’s plasma membrane, immature, so agranular uNK cells are not identified by this reagent[2]. Dolichos biflorus lectin (DBA) is a lectin that specifically binds to terminal N-acetylga lactosamine (GalNAc). DBA lectin is a very selective and sensitive reagent for discriminating mouse uNK cells and is the only leukocyte population in the uterus that binds to DBA[3, 4]. DBA reactions which stain not only the cytoplasmic granules but also the uNK cell membranes have been widely adopted. No lymphocytes in any tissues of virgin mice or external to the uterus of pregnant mice have DBA lectin reactivity equivalent to that of uNK cells. Due to its identification of immature, agranular uNK cells histologically, its usefulness in isolation of uNK cells for RNA isolation when conjugated to immunomagnetic beads and its value in flow cytometry when directly or indirectly conjugated to fluorochromes[5] .Using dual reagent histochemistry with DBA lectin and PAS, all mouse uNK cells are identified [1, 2, 6]. Therefore, in the present study, we focused our attention on the effect of ovarian stimulation on PAS+DBA+ uNK cells. In the revised version, we also added this part to the introduction.
REFERENCES
- Chen, Z.; Zhang, J.; Hatta, K.; Lima, P.D.; Yadi, H.; Colucci, F.; Yamada, A.T.; Croy, B.A., DBA-lectin reactivity defines mouse uterine natural killer cell subsets with biased gene expression. Biol Reprod 2012, 87, (4), 81.
- Sojka, D.K.; Yang, L.; Yokoyama, W.M., Uterine Natural Killer Cells. Front Immunol 2019, 10, 960.
- Paffaro, V.A., Jr.; Bizinotto, M.C.; Joazeiro, P.P.; Yamada, A.T., Subset classification of mouse uterine natural killer cells by DBA lectin reactivity. Placenta 2003, 24, (5), 479-88.
- Yadi, H.; Burke, S.; Madeja, Z.; Hemberger, M.; Moffett, A.; Colucci, F., Unique receptor repertoire in mouse uterine NK cells. J Immunol 2008, 181, (9), 6140-7.
- Croy, B.A.; Zhang, J.H.; Tayade, C.; Colucci, F.; Yadi, H.; Yamada, A.T., Analysis of uterine natural killer cells in mice. 2010.
- Zhang, J.H.; Yamada, A.T.; Croy, B.A., DBA-lectin reactivity defines natural killer cells that have homed to mouse decidua. Placenta 2009, 30, (11), 968-73.
Reviewer 2 Report
The authors report the effect of ovarian stimulation on the cellular and molecular status of the placenta using a mouse model. The subject is unique and important. The manuscript is well-written however many concerns were revealed.
1.In the title add information that the study is performed in mouse placenta
2. In the Abstract only general information is presented, no methods used were listed, and no results were provided
3. Describe how ovarian stimulation in proceeded in the human clinic, similarly add full information for this process in the methodology of your work
4. How many mice were used in total for the study?
5. Provide concentration of used antibodies
6. It will be nice to have estrogen concentration of used for analyses placenta
7. The placental morphology microphotographs are too small
8. It will be nice to have microphotographs of in vitro decidualization of mouse endometrial stromal cells
9. Schematic drawing should be provided showing how studied by you mechanisms are involved in placenta decidualization
Author Response
We appreciate very much your careful review, constructive comments, and kind corrections to our manuscript "ijms-2247003: Ovarian stimulation induced reduction of uterine NK cells proliferation and production of cytokines by perturbing estrogen signaling in mice". We have carefully revised the manuscript according to your comments, and we have incorporated all corrections marked with underlined RED font in the revised version. We hope that we have made some progress in this version. Detailed responses to your questions are given below.
The authors report the effect of ovarian stimulation on the cellular and molecular status of the placenta using a mouse model. The subject is unique and important. The manuscript is well-written however many concerns were revealed.
Point 1: In the title add information that the study is performed in mouse placenta
Response 1: Thank you very much for your advice. We have added information that the study is performed in mouse placenta. The new title is “Ovarian stimulation resulted abnormal placentation by affecting uterine NK cells proliferation and production of cytokines in mice”.
Point 2: In the Abstract only general information is presented, no methods used were listed, and no results were provided
Response 2: Thank you very much for your advice. We have provided methods and results in the abstract.
Point 3: Describe how ovarian stimulation in proceeded in the human clinic, similarly add full information for this process in the methodology of your work
Response 3: Thank you very much for your advice. In human assisted reproductive technology, ovarian stimulation protocols are using follicle-stimulating hormone (FSH) in combination with gonadotropin-releasing hormone (GnRH) (agonists or antagonists), oral supplements[1]. Specific ovarian stimulation programs vary from individual to individual. Ovarian stimulation programs are generally divided into three types: gonadotropin-releasing hormone long program, gonadotropin-releasing hormone short program, and antagonist program[2, 3]. Ovary stimulation commonly involves the use of human chorionic gonadotrophin (hCG): multi-follicular development is induced using exogenous gonadotrophins, and hCG is injected in order to trigger oocyte maturation before retrieving oocytes to perform in vitro fertilization (IVF) or intracytoplasmic sperm injection (ICSI). In the mouse model, the general protocol of ovarian stimulation is that female mice were intraperitoneally injected with pregnant mare's serum gonadotrophin (PMSG) followed by the same international units of hCG 48 h later[4-6]. Therefore, in the present study, the females in the SO group were superovulated by intraperitoneal injection of 5 international units (IU) of pregnant mare serum gonadotropin (PMSG) followed by 5 IU of human chorionic gonadotropin (hCG) 48 h later. The control group was given the same amount of PBS. Full information for this process in the methodology of our work is described in Materials and Methods 4.2 Embryo Transfer in our manuscript. Also, we have described how ovarian stimulation in proceeded in the human clinic in the discussion to the manuscript. We hope that we have made some progress in this version.
Point 4 How many mice were used in total for the study?
Response 4: Thank you very much for your question. About 240 female mice and 25 male mice were used in the study. The above information has been reflected in the manuscript.
Point 5: Provide concentration of used antibodies
Response 5: Thank you for reminding! The concentration of used antibodies has been shown in the manuscript.
Point 6: It will be nice to have estrogen concentration of used for analyses placenta
Response 6: Thank you very much for your advice. I'm sorry that we didn't measure estrogen concentration in the placenta, we used estrogen receptors to measure estrogen concentration. It has been reported in many literatures that estrogen acts through estrogen receptors. After embryo implantation, the endometrium undergoes a decidual response (called decidualization), in which the stromal cells proliferate and differentiate into decidua[7]. The decidual cells surrounding embryos provide nutrients and support for the developing fetus before the placenta starts to fully function. The placenta forms on the mesometrial pole of the uterus, where the blood vessels are supplied via the uterine broad ligament. The implantation and decidualization processes are orchestrated by ovarian steroid hormones through their cognate receptors[8]. In rodents and mammals, ER-a is the predominant estrogen receptor in both the myometrium and the endometrium of uterine tissue[9]. In assisted reproduction treatment, multifollicular development is induced by hormonal stimulation to increase the number of embryos available for transfer. Ovarian stimulation inevitably increases the circulating serum E2 level in the patients[10]. The endometrial stromal cell is the first cell type that comes in contact with circulating E2 in endometrial capillaries and therefore represents the first line of response to E2. E2 binds to ER-a, resulting in receptor dimerization and nuclear translocation. The complex then binds to DNA target sequences to regulate gene expression[9]. We have added the above information to the manuscript. We hope that we have made some progress in this version.
Point 7: The placental morphology microphotographs are too small
Response 7: Thank you very much for your advice. We have changed the placental morphology microphotographs as shown in Figure 1.
Point 8: It will be nice to have microphotographs of in vitro decidualization of mouse endometrial stromal cells
Response 8: Thank you very much for your advice. In early pregnancy, endometrial stromal cells (ESCs) undergo extensive proliferation and differentiation and transform into secretory decidual stromal cells, leading to decidualized endometrium[11]. Decidualization is a critical event for the blastocyst implantation, placental development and fetal growth and the normal term. In mice, the embryo implantation to the uterine epithelial would trigger the endometrial stromal cells to differentiate into decidual stromal cells[12]. Improper decidualization can trigger pathological changes and lead to adverse pregnancy outcomes[13, 14]. We are sorry that we didn’t take micrographs in vitro decidualization of mouse endometrial stromal cells, but the purity of isolated endometrial stromal cells was determined by immunofluorescence staining of vimentin and cytokeratin. Over 95% of the isolated cells were identified as vimentin-positive and cytokeratin-negative stromal cells. We also validated the in vitro decidualization by detecting the expression of dPRP, a well-known decidualization marker of endometrial stromal cells[15]. The expression of dPRP was intensely up-regulated when estrogen and progesterone were added (10nM E2和1μM P4). All the results have been shown in Supplementary Material Figure S1 and Figure S2. The primer sequences for the gene analyzed have been added in Supplementary Material Table S1. In the revised version, we also added this part to the manuscript.
Point 9: Schematic drawing should be provided showing how studied by your mechanisms are involved in placenta decidualization
Response 9: Thank you very much for your advice. We have added schematic drawing to the manuscript showing how studied by our mechanisms are involved in placenta decidualization.
REFERENCES
- Imudia, A.N.; Awonuga, A.O.; Doyle, J.O.; Kaimal, A.J.; Wright, D.L.; Toth, T.L.; Styer, A.K., Peak serum estradiol level during controlled ovarian hyperstimulation is associated with increased risk of small for gestational age and preeclampsia in singleton pregnancies after in vitro fertilization. Fertil Steril 2012, 97, (6), 1374-9.
- Farhi, J.; Ben-Haroush, A.; Andrawus, N.; Pinkas, H.; Sapir, O.; Fisch, B.; Ashkenazi, J., High serum oestradiol concentrations in IVF cycles increase the risk of pregnancy complications related to abnormal placentation. Reprod Biomed Online 2010, 21, (3), 331-7.
- Styer, A.K.; Wright, D.L.; Wolkovich, A.M.; Veiga, C.; Toth, T.L., Single-blastocyst transfer decreases twin gestation without affecting pregnancy outcome. Fertil Steril 2008, 89, (6), 1702-8.
- Mainigi, M.A.; Olalere, D.; Burd, I.; Sapienza, C.; Bartolomei, M.; Coutifaris, C., Peri-implantation hormonal milieu: elucidating mechanisms of abnormal placentation and fetal growth. Biol Reprod 2014, 90, (2), 26.
- Fortier, A.L.; Lopes, F.L.; Darricarrere, N.; Martel, J.; Trasler, J.M., Superovulation alters the expression of imprinted genes in the midgestation mouse placenta. Hum Mol Genet 2008, 17, (11), 1653-65.
- Ertzeid, G.; Storeng, R., The impact of ovarian stimulation on implantation and fetal development in mice. Hum Reprod 2001, 16, (2), 221-5.
- Ramathal, C.Y.; Bagchi, I.C.; Taylor, R.N.; Bagchi, M.K., Endometrial decidualization: of mice and men. Semin Reprod Med 2010, 28, (1), 17-26.
- Winuthayanon, W.; Lierz, S.L.; Delarosa, K.C.; Sampels, S.R.; Donoghue, L.J.; Hewitt, S.C.; Korach, K.S., Juxtacrine Activity of Estrogen Receptor alpha in Uterine Stromal Cells is Necessary for Estrogen-Induced Epithelial Cell Proliferation. Sci Rep 2017, 7, (1), 8377.
- Yilmaz, B.D.; Sison, C.A.M.; Yildiz, S.; Miyazaki, K.; Coon, V.J.; Yin, P.; Bulun, S.E., Genome-wide estrogen receptor-alpha binding and action in human endometrial stromal cells. F S Sci 2020, 1, (1), 59-66.
- Chai, J.; Lee, K.F.; Ng, E.H.; Yeung, W.S.; Ho, P.C., Ovarian stimulation modulates steroid receptor expression and spheroid attachment in peri-implantation endometria: studies on natural and stimulated cycles. Fertil Steril 2011, 96, (3), 764-8.
- Wang, J.; Tang, Y.; Wang, S.; Cui, L.; Li, D.; Du, M., Norepinephrine exposure restrains endometrial decidualization during early pregnancy. J Endocrinol 2021, 248, (3), 277-288.
- Liu, H.; Huang, X.; Mor, G.; Liao, A., Epigenetic modifications working in the decidualization and endometrial receptivity. Cell Mol Life Sci 2020, 77, (11), 2091-2101.
- Chen, W.; Lu, S.; Yang, C.; Li, N.; Chen, X.; He, J.; Liu, X.; Ding, Y.; Tong, C.; Peng, C., et al., Hyperinsulinemia restrains endometrial angiogenesis during decidualization in early pregnancy. J Endocrinol 2019, 243, (2), 137-148.
- Zheng, H.T.; Fu, T.; Zhang, H.Y.; Yang, Z.S.; Zheng, Z.H.; Yang, Z.M., Progesterone-regulated Hsd11b2 as a barrier to balance mouse uterine corticosterone. J Endocrinol 2020, 244, (1), 177-187.
- Rasmussen, C.A.; Orwig, K.E.; Vellucci, S.; Soares, M.J., Dual expression of prolactin-related protein in decidua and trophoblast tissues during pregnancy in rats. Biol Reprod 1997, 56, (3), 647-54.
Round 2
Reviewer 1 Report
I propose that this article, after eliminating the comments and correcting minor errors, may be published.
Author Response
No
Reviewer 2 Report
The study was improved according suggestions
Author Response
No